# Microbiomes, Epigenomics, Immune Response, and Splicing Signatures Interplay: Potential Use of Combination of Regulatory Pathways as Targets for Malignant Mesothelioma

**DOI:** 10.3390/ijms23168991

**Published:** 2022-08-12

**Authors:** Botle Precious Setlai, Zilungile Lynette Mkhize-Kwitshana, Ravi Mehrotra, Thanyani Victor Mulaudzi, Zodwa Dlamini

**Affiliations:** 1Department of Surgery, Level 7, Bridge E, Steve Biko Academic Hospital, Faculty of Health Sciences, University of Pretoria, Private Bag X323, Pretoria 0007, South Africa; 2Department of Medical Microbiology, School of Laboratory Medicine & Medical Sciences, Medical School Campus, College of Health Sciences, University of KwaZulu-Natal-Natal, Durban 4041, South Africa; 3India Cancer Research Consortium (ICMR-DHR), Department of Health Research, Red Cross Road, New Delhi 110001, India; 4SAMRC Precision Oncology Research Unit (PORU), DSI/NRF SARChI Chair in Precision Oncology and Cancer Prevention (POCP), Pan African Cancer Research Institute (PACRI), University of Pretoria, Hatfield 0028, South Africa

**Keywords:** mesothelioma, epigenetics, MicroRNA, microbiome, immune modulation, alternative splicing, asbestos, therapeutic targets

## Abstract

Malignant mesotheliomas (MM) are hard to treat malignancies with poor prognosis and high mortality rates. This cancer is highly misdiagnosed in Sub-Saharan African countries. According to literature, the incidence of MM is likely to increase particularly in low-middle-income countries (LMICs). The burden of asbestos-induced diseases was estimated to be about 231,000 per annum. Lack of awareness and implementation of regulatory frameworks to control exposure to asbestos fibers contributes to the expected increase. Exposure to asbestos fibers can lead to cancer initiation by several mechanisms. Asbestos-induced epigenetic modifications of gene expression machinery and non-coding RNAs promote cancer initiation and progression. Furthermore, microbiome–epigenetic interactions control the innate and adaptive immunity causing exacerbation of cancer progression and therapeutic resistance. This review discusses epigenetic mechanisms with more focus on miRNAs and their interaction with the microbiome. The potential use of epigenetic alterations and microbiota as specific biomarkers to aid in the early detection and/or development of therapeutic targets is explored. The advancement of combinatorial therapies to prolong overall patient survival or possible eradication of MM especially if it is detected early is discussed.

## 1. Introduction

Malignant mesothelioma (MM) affects the mesothelium cell lining. These include the lining around the heart, the tunica vaginalis testis, the lungs, and the abdomen. There are three types of mesothelioma, namely the epithelioid, sarcomatoid, and biphasic. Malignant mesothelioma is mainly caused by exposure to asbestos fibers. It is considered a rare and deadly disease [1,2] with median overall survival of 9–18 months [3]. The incidence of MM has had a worldwide gradual increase over the past years and is predicted to be at its highest in 2020 [4]. However, the disease is highly misdiagnosed in Sub-Saharan African (SSA) countries. A significant number of SSA communities still use asbestos products and most of these are housing structures. Concerns around exposure to asbestos fibers are mainly associated with the mining industry [5]. This notion might be skewed as some of the small community members who do not work nor reside near the mines have succumbed to mesothelioma or some kind of undiagnosed lung disease.

The CpG methylation profiles were identified as independent predictors of patient survival. Epigenetic alterations observed in cancers vs. non-cancerous samples correlated with asbestos body burden, providing evidence that asbestos exposure characterizes tumors into specific epigenetic subclasses. Irregular epigenetic mechanisms are shown to be responsible for the induction of malignant pleural mesothelioma (MPM) [6]. DNA methylation is a well-studied epigenetic modification in cancer [7,8]. This modification is responsible for guiding cell characterization through gene expression, genome stability, or blocking the interaction between DNA and related transcription factors [9,10]. Cancer cells take advantage of dysregulated DNA methylation by targeting CpG islands in gene expression regulatory machinery [10].

MicroRNA (miRNA) are non-coding RNAs in which their expression is in part regulated by other epigenetic features, such as DNA methylation and histone modifications [11]. It has long been established that miRNA expression is dysregulated in mesothelioma [12] and these miRNAs target cell-cycle transcription factors whilst being regulated by the components of the cell cycle themselves. These features reflect an indication of their importance as potential therapeutic biomarkers of diseases such as MM [13]. The other well-studied key regulator of the immune system is the microbiome. The microbiome assists in the development and maturation of the immune system. The balance between the host immune system and microbiome contributes immensely to susceptibility to inflammatory diseases (bearing in mind that cancer is considered a chronic inflammatory disease) later in life. The gut microbiome has been shown to have an impact on cancer immunosurveillance [14,15], and this impact is mainly attributed to microbiota in mesotheliomas [16]. Thus, it is important to understand how epigenetic modifications and interactions with microbiome and related immune responses could provide potential therapeutic targets for the early diagnosis and treatment of MM.

## 2. The Burden of Asbestos-Related Diseases, a Persisting Challenge

The global incidence of mesothelioma increased during the period of 1990 to 2017. Over 50% of the cases recorded were from high socio-demographic index (SDI) regions. The age-standardized incidence rate (ASIR) decreased from 1.11 to 0.17% in Southern SSA after 2000 but increased from 2.03 to 2.30% in Australasia. In this study, Zhai et al. reported that the global trends of mesothelioma vary amongst countries, but according to the authors, the incidence of mesothelioma has decreased since 1990 [17]. Screening and early detection of mesotheliomas remains a challenge; hence, the incidence of the disease is not well recorded. The lack of resources in the LMICs could be a limiting factor in terms of diagnosing and recording mesothelioma incidences and mortality rates as expected.

Recently, Chimed-Ochir et al. evaluated the correlation of country-level mesothelioma burden and asbestos use with national income status. The study looked at 80 high-income countries of which 54 (68%) reported mesothelioma deaths to WHO. The low-middle-income countries (LMICs) were 78, and only 11 (14%) of these countries reported mesothelioma deaths. The other 86% of these deaths were not reported. The highest number of mesothelioma deaths were recorded by high-income and upper-middle-income countries at 29,854 (78%), whilst LMICs reported only 8534 (22%) deaths [18]. This study echoes the need for social and scientific community awareness campaigns so that the communities are aware of the dangers of occupational and household asbestos products. The companies that still produce products that contain asbestos need to be evaluated by health professionals and inspected regularly. The need to facilitate the implementation of the regulations to control asbestos exposure should be considered. Finally, simpler reliable screening methods that can be utilized even in rural areas for early detection of asbestos-related diseases should be ventured into.

In LMICs such as India, the continual use of asbestos products is remarkable. Jadhav and Gawde, 2019 predict that the country will experience a high incidence of asbestos-related diseases accounting for at least 1.25 million patients diagnosed with cancer worldwide [19]. The latency of mesothelioma is long and highly variable with a range of 13–70 years meaning that although the disease is considered rare, there is a chance that the numbers are yet to increase among people who have been exposed to asbestos before the ban or implementation of regulatory measures. The socioeconomic status in rural SSA regions makes the decision to eliminate the use of asbestos difficult as transitioning from asbestos roofing to metal sheets, for example, can be costly. Owing to its long, covert pathogenesis and latency, scientists in the SSA regions do not pay much attention to the disease as it is not necessarily obvious like other cancers commonly diagnosed in the region. There is thus the need to find better ways of diagnosing the disease with consideration of the asbestos exposure situation in LMICs like SSA and India in mind. There is a possibility that more and more cases of mesothelioma will emerge. There will have to be measures implemented to ensure that these cases are not missed and efforts to improve the quality of life are undertaken.

Wagner et al. reported cases of mesothelioma induced by exposure to asbestos originating from the mines. However, one of the cases was peculiar as the patient had never worked at the mines. The relatives of the patient also mentioned no previous knowledge of asbestos exposure neither through the mining industry nor factories manufacturing asbestos production [20]. Although it is a major problem, repurposing of asbestos products for household purposes in these communities is still a normal practice. For example, asbestos sheets can be used to dry animal skin or meat prepared for traditional ceremonies. The manufacturing of asbestos products used to be a lucrative business for South Africa. The discovery of its detrimental effects on human health led to the ban of the production of asbestos products in the country. It is important to note that there is still a substantial number of houses with asbestos cement roof sheets. Guidelines for the demolition of these structures have been set by the National Institute for Occupational Health (NIOH) of South Africa. The NIOH also analyzed the samples removed from the structures and measures the asbestos fibers post demolition. Although not all of the structures were demolished, 2990 samples were collected from 3/9 provinces in the country (Gauteng, Mpumalanga, and the Western Cape). A total of 1581 bulk samples were collected. The study found that 54.8% of the bulk samples contained asbestos, and 16.1% (227) of asbestos was found in air filters post demolition [21]. This is a potential indication of how asbestos fibers escape into the air during demolition; hence, the area of demolition must be well contained.

Efforts to prevent asbestos-related diseases require implementation of a set of sequential measures. Ideally, this would include managing and removing the existing asbestos housing structures and asbestos-related products replacing them with alternative materials such as iron sheet roofing; exposure to occupational asbestos fibers can be limited by taking preventative measures such as training employees on asbestos exposure and related risks; in cases where exposure is unavoidable, training on how to safely handle asbestos and related products should be implemented; the use of personal protective equipment (PPE) where necessary should be encouraged; regular measurements of asbestos fibers in the air should be done. Perkins et al. reiterated the significance of enough wetting during the demolition processes as it assists with the reduction of airborne materials including asbestos fibers [22].

## 3. The Role of miRNA in Mesothelioma

The miRNAs involved in oncogenesis pathways are reduced in MM encouraging the development of miRNAs mimics for possible reversal of cancer initiation processes [23]. Studies done on experimental models have demonstrated the possibility of reversing oncogenesis by reintroducing specific miRNAs identified as tumorigenic. For instance, treatment of MPM tumor xenografts with ectopically re-expressed miR-206 showed a significant reduction of tumor growth achieved via G1/S cell cycle arrest. The effectiveness of miR-206 was shown to be controlled by the RTK-Ras-MAPK-PI3K/Akt-CDK pathway. This effect allowed the authors to ascertain the use of CDK6 as a novel target for miR-206 [24]. Similarly, treatment with miRNA-215-5p diminished cancer growth by activation of mouse double minute 2 (MDM2)-p53 signaling pathway and resultant caspase-dependent apoptosis [25]. The miR-126 located within intron 7 of its host gene EGFL7 is downregulated in MPM. The EGFL7 S2 region methylation status was associated with significantly worse MPM patient survival. These results correlated well with the downregulation of EGFL7 transcript variant 1 and miR-126 [26]. Significant downregulation of miR-126 is strongly correlated with serum levels of vascular endothelial growth factor (VEGF) [27] and soluble mesothelin-related peptide (SMRP). SMRP was previously identified as a possible diagnostic marker for MPM [28]. Follow-up studies identified pleural effusion SMRP as a better indicator of MPM over serum SMRP [29]. The MDM2 along with other factors such as TRAIL [30]/HIF-1α [31] involved in MPM have been investigated as therapeutic targets or potential biomarkers, respectively. Identification of miRNA as potential therapeutic targets of MM has been a topic of interest for a while, these include miR-16 [32], miR-193a-3p [33], miR-17-5p in relation to KCa1.1 [34], and miR-411, which controls the expression of IL-18 [35] (Table 1). Targeting miRNAs and related pathways continue to be the topic of interest today.

## 4. Epi-Regulation of miRNAs by Methylation Processes

The majority of eukaryotic genome modifications resulting in downregulation of gene expression, including that of miRNAs, are ascribable to DNA methylation. This takes place at the CpG islands located in the proximal promoter regions of most genes. During embryonic development, DNA methylation guides specific lineages but, hypermethylation of CpG islands can result in silencing of tumor suppressor genes. Tumor suppressor miRNAs can be silenced by dysregulated methylation of CpG islands adjacent to their promoters [48]. The concept of the regulator becoming regulated was discussed by Wang et al. While DNA methylation can occur in whole-genome modifications, it can also take place at the promoter regions and impact the transcription of genes relevant for cancer development and progression. The miRNAs on the other hand have a negative impact on gene expression, which could manifest in diseases such as cancer. The miRNAs control DNA methylation through their interaction with methyltransferases [49]. These miRNAs referred to as epi-miRNA control the expression of enzymes responsible for DNA methylation as well as histone modifications, thus affecting the overall epigenome. The aforementioned epigenetic modifications are the main perpetrators of the miRNA dysregulation in cancer [50]. In return, hypo/hypermethylation of miRNAs decides the fate of miRNA products as either inducers of tumorigenesis or tumor suppressor genes [49].

## 5. Epigenetic miRNA as Potential Diagnostic Biomarkers and Targeted Therapies

The DNA methylation processes as modulators of miRNA expression have been investigated and shown to be a possible reliable diagnostic or therapeutic tools. Epigenetic modifications are reversible; thus, identifying and targeting specific miRNAs to reverse the effects of asbestos related diseases might be of great significance [51]. The miRNAs such as miR-486-5p are downregulated in MPM. Re-expression of miR-486-5p results in sensitivity to chemotherapeutic treatment by advancing apoptosis and disruption of mitochondrial function in cancer cells [23]. The miR-486-5p is epi-regulated by cancer-related methylation of the ANK1 variant 1–4 in which it is located. Hypermethylation takes place at the CpG island upstream of miR-486 and contributes to its possible role in halting or inhibiting cancer progression [52].

The potential use of miRNA as a diagnostic tool for asbestos exposure and MM was assessed by Micolucci et al. The miRNAs with biomarker potential for MM were denoted as mesomiRNAs. The study found that circulating mesomiRNAs can be used in combination with mesothelin as potential markers of asbestos-exposed MM [53]. Subunits of miR-34 are transcription targets for the tumor suppressor p53. Deregulated methylation was observed in 85% of MPM tumors with miR-34b/c expression. The expression of miR-34a/b/c, which is downregulated by methylation in MM cell lines, was normalized by treatment with an epigenetic drug, decitabine. Transfection of MPM cells expressing miR-34b/c showed a reduction in cancer proliferation and migration [41]. In agreement with this study, miR-34b/c was downregulated by DNA methylation in ~90% of MPM cases. Methylation was more pronounced in advanced diseases, which had higher expression of miR-34b/c than early MPM cases [42]. The in vivo studies with an adenovirus vector expressing miR-34b/c significantly reduced tumor growth, thus suggesting possible use as an effective treatment for MPM [43]. The latest study done on the assessment of methylated miR-34b/c suggests the use of droplet digital polymerase chain reaction (ddPCR) as a reliable method for the detection of methylation in circulating DNA of MPM patients [44], as shown in Figure 1.

## 6. Epigenetic miRNA in Other Asbestos-Related Diseases, Possible Relation to MM

Generally, exposure to asbestos fibers can induce epigenetic alterations that silence tumor suppressor genes [54]. These epigenetic or genomic changes are linked to reactive oxidative stress (ROS) and cancer initiation. Asbestos fibers can pass through the lining of the lungs. Tissue macrophages will then be activated and try to get rid of these fibers via phagocytosis releasing a large number of ROS. Increased concentration of ROS induces aberrant molecular processes that lead to DNA damage. This results in the transformation of mesothelial cells into cancerous cells. In the process, exposure to asbestos fibers induces dysregulated miRNA expression [55,56] (Figure 2). Specific miRNAs such as miR-197-3p were found to be significantly downregulated in individuals who were exposed to work-related asbestos than controls. The authors suggested the use of miR-197-3p as a potential biomarker of asbestos exposure [57]. Earlier Tomassetti et al. noted the importance of early detection in the possible reduction of cancer burden and how this could assist in treatment efficacy. The common method for measuring asbestos exposure is by chest X-ray. This method is costly and time consuming, especially in resource-limited countries where the waiting list can be long. The study suggested the use of circulating epi-nuclei acids as predictive biomarkers intended for therapeutic purposes [58].

Analysis of samples of patients with asbestos-induced lung cancer and mesothelioma identified miRNAs significantly associated with malignancy. The miR-205 was identified as specific for non-small cell lung cancer (NSCLC) only, miR-520g was specific for asbestos-related NSCLC, and the miR-222 and miR-126 were identified as specific for MPM [59]. The risk of developing asbestos-induced lung cancer can be assessed through DNA methylation. The resultant alterations in gene expression, DNA copy number, and miRNA profiles have been reported [54]. A set of mesothelioma cells transfected with miR-1 and miR-214 lost their ability to proliferate and metastasize. The flow cytometry results revealed a cell cycle arrest at the S and G2/M phases [60]. Both miR-1 [61] and miR-214 [62,63] were silenced by epigenetic mechanisms during the development and cancer proliferation. The miR-1 involves the initiation of hepatocellular carcinoma whilst miR-214 contributes to drug resistance in renal cell carcinoma and pediatric intracranial non-germinomatous malignant germ cell tumors. Thus, this set of miRNAs could be considered as alternative treatment strategies for MM or in combinatorial therapies to decipher drug resistance.

## 7. The Connection between Epi-miRNA and Breast Cancer Gene 1-Associated Protein 1

The most commonly identified germline mutation in mesothelioma is breast cancer gene 1-associated protein 1 (BAP1). The BAP1 is a ubiquitin carboxy-terminal hydrolase that functions as a tumor suppressor gene. It serves as one of the preeminent proteins that are involved in the modulation of the cell cycle, cellular differentiation, and DNA damage response [64]. Individuals with BAP1 gene mutation tend to develop an inherited disorder known as the BAP1 tumor predisposition syndrome, which makes them susceptible to various cancers including MM [65]. The BAP1 mutations are present in about 60% of MPM cases and are associated with improved overall survival [66]. In agreement with the previous author, BAP1 was detected by immunohistochemistry in 8 out of 22 tissue samples with wild-type BAP1, and none was identified in 14 of the tissue samples with the BAP1 mutation. The overall results of 70 MM tissue samples studied revealed that the BAP1 gene could not be detected in 67.1% of these samples. These results correlated well with molecular studies indicating the consistency between the two methods of analysis [67].

The BAP1 mutations have a lower tumor mutation burden associated with poor prognosis, which worsens when it is considered in combination with mucin (MUC) 16 [68]. Some of the mucin family of proteins form a physical barrier that serves as the first line of defense for the epithelial layer surrounding the respiratory and gastrointestinal organs [69], a target for MM. Mesothelioma can be difficult to diagnose due to its histopathological features, thus finding ways to distinguish MM from other pleural cancers is crucial. This is more evident in mucin-positive epithelial mesothelioma, which has misleading morphological features [70]. The MUC4 protein was used to differentiate pleural mesothelioma from lung (pleural) adenocarcinoma and was found to be a more significant marker of the latter disease [71]. However, it should be noted that there are other non-BAP1 mutations associated with cancer predisposition in high-risk cancer families with MM. Predicted pathogenic mutations include alterations in the DNA repair or chromatin modification, with the most prominent mutation being the loss of leucine-rich repeat kinase 2 (LRRK2) expression [72].

Downregulation of miR-31 was observed in epithelioid MM versus non-epithelioid MM. Patients with BAP1 loss/low miR-31 combination had a better prognosis; hence, this was seen as an independent prognostic factor for epithelioid-MPM patients [47]. The BAP1 mutation-specific miRNA signature has previously been identified as prognostic biomarkers in other cancers as well [73] whilst BAP1 alone could not be correlated with clinical outcomes [74,75]. A total of 11 miRNA identified were linked with BAP1. Eight of these miRNAs—miR-149, miR-29b-2, miR-182, miR-183, miR-21, miR-365-2, miR-671, and miR-365-1—were associated with worst overall patient survival, and miR-10b, miR-139, and miR-181a-2 were associated with improved overall survival. This miRNA signature was considered an independent prognostic factor for BAP1 wild-type cancer [73]. It could be worthwhile to assess the impact of the same array of miRNA in other BAP1 mutated cancers such as MM as well. BAP-1 has been shown to modulate the expression of miRNA and related proteins. The BAP1 mutations interfere with BAP1–BRCA–BARD1 protein interaction and result in a conformational transition that affects the binding affinity of miRNAs to the BAP1 locus (Figure 3), ultimately modulating their regulatory networks [76].

Mutations in the BAP1 gene can also interfere with DNA transcription resulting in modulation of the miRNA product. miR-31, used as an example here, is associated with better patient prognosis in MM. The miR-31 expression is downregulated in MM. This can be influenced by mutated BAP1 through alterations in transcription processes as indicated by purple arrows, ultimately affecting translation. Normal BAP1 by uniform round shape in brown color and mutated BAP1 is depicted in a disfigured oval shape in the same color.

### The Role of BAP1 in Immune Response and How It Relates to the Current Treatment Modalities

Aberrant BAP1 expression along with immune modulation is significantly associated with reduced patient survival. The diversity of tumor-infiltrating immune cells can be determined by the tumor phenotype and aberrations in the BAP1 gene [71]. Inhibition of chemokine receptor 5 (CCR5) in tumors with BAP1 mutation improved cytotoxicity and antigen presentation activity by dendritic cells. The effect was accompanied by the inhibition of immune checkpoint inhibitors (ICs) [77], which advocates for cancer cells survival and progression in MM. Treatment of MM has been formidable leading to several studies including clinical trials conducted to find effective treatment modalities for the disease. The Food and Drug Administration (FDA) recently approved a combination of ipilimumab and nivolumab as the primary treatment option for MPM that cannot be treated by surgical intervention [78]. The clinical outcome of BAP1 mutation in MPM patients treated with immunotherapy was studied by Dudnik et al. The study found that treatment response to PD-1/PD-L1 inhibitors was not defined by the presence of BAP1 mutations [79]. However, a combination of BAP1 mutations, mucin proteins, and immunoregulatory factors may be of value. For instance, MUC1 in combination with HLA-A*02, part of the immune regulatory complex with antigen-presenting capacity has been recognized as potential immunotherapeutic targets for MPM [80]. The loss of BAP1 can upregulate the expression of immunological genes, such as HLA-DR, HLA class II, and CD38, correlated with immunosuppressive cells infiltrating the tumor microenvironment [81].

The involvement of immunosuppressive cells and related signaling pathways threatens the effective development of anti-cancer drugs and overall patient survival. These immune cells include the suppressive cells, such as M2-like/tumor-associated macrophages (TAMs) and regulatory T cells (Tregs), known to favor and facilitate cancer progression [82]. Loss in BAP1 was associated with epigenetic regulation of PROS1. The PROS1 gene is involved in phosphorylation and activation of the immunosuppressive macrophage receptor, MERTK. The MERTK is correlated with increased TAMs marker CD163 [83]. These data agree with the work done by Figueiredo et al. Their study found a correlation between BAP1 mutation and immunosuppressive genes HLA-DR, CD38, and CD74. The most predominant tumor-associated cells were CD8+ Tregs and TAMs [81].

The quantity of B cells lineage was significantly reduced in mice with the loss of BAP1 compared to controls. A reduction in the number of B cells in the spleen and the bone marrow of BAP1-mutated mice was observed. The loss of BAP1 affected B cells lineage development by suppressing cell cycle progression in premature, immature, and mature B cells. B cells play an important role in anti-tumor immunity; thus, the study concurs with previous ones that indicted that BAP1 mutations can have favorable effects on cancer progression [84]. A reduction in B cells was also observed in BAP1-mutated tumors in MPM. This was concurrent with a decrease in NK cells and an increase in mast and CD3+CD8+ T cells [85]. Impaired B cells development was observed in mice treated with tamoxifen. Chemotherapeutic treatment of mice with tamoxifen resulted in the loss of the BAP1 gene. BAP1 is needed for some of the stages of T cells development. The deletion of BAP1 affects the differentiation of thymocytes into all subsequent immature cells. Consequently, these mice had severe thymic atrophy accompanied by reduction in γδ T cells. Molecular analysis of these cells showed defects in E2Fs target genes, which are critical regulators of the cell cycle. Although CD4+CD8+ T cells were not reduced, BAP1 was still required for their maintenance. Peripheral neutrophils and monocytes were also increased [86].

## 8. Epigenetic and Immune Modulation as Therapeutic Strategies

The idea of combining cancer immunotherapy with epigenetic modulators has always been an attractive one. The ability of cancer cells to develop ways to use components of the immune system in their favor and avoid immunosurveillance articulates the need for the development of other therapeutic interventions, such as histone deacetylases or DNA methyltransferases inhibitory drugs. These drugs can modulate the immunoregulatory system and heighten response to anti-PD-1 immunotherapy [87]. Based on the notion that epigenetic factors can modulate immunoregulatory-related genes, Anichini et al. mapped the landscape of these genes and found that DNA methyltransferase inhibitors robustly induce innate immunity pathways. Epigenetic drugs regulated genes encoding immune-related genes; HLA class I and HLA class II and IFN, TNF, and TGF-β pathways (involved in both anti- and pro-inflammatory responses) with guadecitabine being the most effective [88].

Similar results were observed in a study performed on mesothelioma cell lines. The DNA hypomethylating agent, guadecitabine, upregulated the expression of HLA class I antigens, PD-L1, and natural killer group 2-member D Ligands (NKG2DLs). The effects were pronounced with the addition of histone deacetylases inhibitors and EZH2 inhibitors. The authors suggested the use of a combination of DHA-based immunotherapies for the treatment of mesotheliomas [89]. Of concern, is the upregulation of PD-L1, which should be inhibited (hence, treatment modalities such as nivolumab) as part of the approved treatment protocol for MPM (vide supra). The ICs can be administered in a form of gene therapy to enhance anti-tumor immune responses. The mPEG-b-PLG/PEI-RT3/DNA polymeric gene delivery system was used as a vehicle for plasmid DNA encoding shPD-L1 to reverse T-cell exhaustion. This system, in combination with zebularine, a DNA methyltransferase inhibitor, inhibited tumor growth and metastasis [90].

## 9. Involvement of Microbiota as Regulators of Immune Response

Microbiota immunomodulation and involvement in cancer development has been discussed extensively in a number of papers [91,92,93]. Microbiota can modulate both the innate and adaptive immune responses as a mechanism to influence treatment response [94]. Thus, this section will focus on the microbiota–immune modulation and therapeutic implications that could be applicable as MM therapeutics. The mainstay mechanism for MM induction is asbestos exposure (vide supra) and the airway microbial network is implicated in allowing for passage of asbestos fibers to enter and penetrate the pleural linings of the respiratory organs. The involvement of microbes such as simian virus 40 (SV40) release secretory molecules, which serve as predispositions for MM [95]. Higuchi et al. identified 36 species of microbiota in MM tissues, with Streptococcus australis and Ralstonia pickettii being the most abundant. This microbiota was identified as mesothelioma-related microbiota, which contributes to cancer progression. The investigators suggested the use of this microbiota as possible therapeutic targets for mesothelioma [16]. Evidence-based knowledge indicates that microbiota-altered immune response provides a direct link to patient response to cancer immunotherapy. The approval of cancer immunotherapy as the primary treatment option for MM, and its lack of efficacy could be attributed to microbial diversity in MM.

The effect of 12 clostridia strains, immunomodulators of Tregs, was assessed to design computational indexes that can be used to identify “immune-modulating bacteriotherapeutics”. This group of microbiota induce Tregs activation [96] and could be targeted to help dampen Tregs function in cancer, which has been shown to influence the therapeutic efficacy of PD-1/PD-L1 inhibition [97]. In a classic mouse model, Wang et al. demonstrated a species-specific role of Bifidobacteria in enhancing the anti-melanoma immune response. The mechanism was associated with the Bifidobacteria-stimulated dendritic cells resulting in enhanced tumor-specific CD8+ cytotoxic T cells function, which increased immune-checkpoint (anti-PD-L1) efficacy [98]. The different immunomodulatory mechanisms exhibited by microbiota, in different cancers as discussed in this section, warrant further, well-structured research into these interactions.

The composition of the gut microbiome modulates the immune system via activation of anti-cancer systems (dendritic cells/memory T cells) or cancer-favoring cells such as Tregs and related cytokines. This makes translating microbial therapeutics from preclinical studies to clinical settings a challenge. Several methods have been suggested to identify specific microbiome that can alter the immune response and epigenetic pathways. Therapeutic solutions to an altered microbial composition might also be fecal transplantation as dysbiosis correction therapy (DCT) [99]. DCT could be valuable when applied in combination with IC inhibitors to improve treatment response in mesotheliomas. The selection of microbiota that can be effectively utilized in the improvement of the patient’s immunological response, and ultimate therapeutic efficacy might be a difficult venture to take on. However, it might add value to the continual efforts to discover the ultimate treatment strategy for cancers that are difficult to treat such as MM.

## 10. Host Microbiome and Epigenetic Regulation

Earlier on we highlighted the role of the respiratory microbiome in the development of mesothelioma and its attributes to drug resistance. Similarly, colorectal cancer has a fair share of microbial diversity by virtue of its primary location. Colorectal cancer can advance by taking advantage of the interaction between the host and microbial consortia. Disruption in host and commensal microbes interplay alters their harmonious co-adaptation leading to dysregulated epigenetic mechanisms. The direct interaction between microbiota or indirect interaction through their secreted metabolites, such as short-chain fatty acids and epigenetic factors, is responsible for the induction of microbial-related epigenetic modulation. The microbiota has been shown to control T cells differentiation via epigenetic modifications (Figure 4). This includes Tregs, which are valuable in the implementation and maintenance of a homeostatic environment accommodative to both commensal microbiota and its host. Disruptions in the epigenetic cross-talk between the host and commensal microbiota can induce or promote cancer progression [100].

The value of the composition of the commensal microbiome in the prevention of diseases is undeniable in cancers such as cervical cancer. Microbial infection with the human papillomavirus (HPV) is well recognized as the major cause of ovarian and cervical malignancies [101]. This species of microbes is even identified as a possible diagnostic marker in breast [102] and head and neck cancers [103]. In breast cancer, for instance, HPV is associated with highly dysregulated methylation in the p97 promoter region of HPV 16 [102]. A cluster of differentially expressed miRNAs was directly correlated with the composition of microbial taxa in colorectal cancer patients’ tissue samples [104]. Alterations in epigenetic mechanisms are affected by molecular factors secreted by the metabolic activity of the host microbiota. The function of methylases and acetylases involved in DNA methylation/histone modifications can be regulated by microbiota metabolites [105]. Microbiota is additional suppliers of methyl and acetyl needed by these epigenetic enzymes to perform their catalytic functions. Metabolic degradation of folate by microbiota produces the main substrate for DNA and histone methylation known as S-adenosylmethionine [106]. Therefore, the data provided here suggest that the relationship between microbiota and epigenetic regulation may prognosticate carcinogenesis in MM. This knowledge can be applied when venturing into the development of efficient therapeutic interventions for MM.

## 11. Epigenetic Alterations of Splicing in MM

Alternative splicing (AS) and epigenetic mechanisms share a common role in terms of gene expression regulatory processes. They have both been intensively studied for their contribution to malignant transformation. Both components of AS and epigenetic machinery are associated with cancer initiation and progression. Of note, components of epigenetic mechanisms, in particular, chromatin and histone modifications, have been reported as key regulators of AS. Nucleosomes (which form the core of the histones) preferentially interact with exons rather than introns. There is a correlation between AS and DNA transcription. RNA polymerase II (Pol II), an enzyme responsible for elongation, prefers positioning over the exons over the introns. This positioning allows the enzyme to recruit the spliceosome needed for AS. Furthermore, chromatin and histone modifications control AS via kinetic coupling, which involves competitive binding of Pol II and spliceosome to initiate elongation and splicing, respectively. Pol II takes precedence and facilitates exon skipping [84], as shown in Figure 4.

The enzyme RNA pol II positions itself over the exons and recruits the spliceosome machinery to initiate AS. In return, the histone and chromatin compartments regulate the function of the AS processes. This can be achieved by RNA pol II competing for binding on the exons preventing the spliceosome from binding in the same position to initiate AS. Viruses take advantage of aberrations in alternative splicing and promote cancer initiation and progression. On the other hand, the presence of microbiota or metabolites secreted by these organisms initiates epigenetic modifications, which control immune responses by inducing T cells differentiation into different subsets. The blue and pink rectangular boxes denote exons, and the purple rods are introns. 

Alterations in alternative splicing (AS) profiles of MPM samples were characterized using RNA sequencing data. Recurrent mutations including that of splicing factor SF3B1 were identified. The alterations in splicing factors were associated with aberrant expression levels of BAP1 discussed in the previous sections of this review. The authors report that alterations in splicing could be one of the extensive mechanisms used in MPM for the downregulation of BAP1. The study also found the levels of PD-L1 to be increased in the tumor microenvironment of MPM [107] correlating with previous studies that lead to the use of anti-PD-L1 as one of the primary treatments of choice for this disease. Indeed, three years later, Sciarrillo et al. studied alterations in splicing as a potential prognostic or therapeutic target in diffuse MPM. The authors note that SF3B1 encodes enzymes involved in epigenetic modifications. These enzymes are involved in RNA processing and splicing processes. To study the possible use of this combination of regulatory molecular mechanisms as an anti-cancer therapy, the authors correlated the expression of splicing factors with clinical outcomes. Upregulation of SF3B1 was associated with significantly worse prognosis and was considered to be an independent marker for disease progression and mortality [108].

Based on the notion that epigenetic modifications can be reversed and aberrant AS features can be corrected, Gimeno-Valiente et al. highlighted the importance of considering the interrelation between epigenetic modifications and alternative splicing as potential therapeutic targets. Both processes have been intensively studied for their contribution to malignant transformation. Aberrations in AS can give rise to mRNA isoforms from the same gene with antagonistic functions. Other mRNA isoforms could be oncogenic while other genes present with tumor-suppressive capabilities. The connection between miRNAs, such as miR-193a-5p and splicing factor SRSF6, is associated with the activation of epithelial-to-mesenchymal transition involved in cancer progression and metastasis. The authors have also indicated that miRNAs targets mRNA isoforms that could either down- or upregulate splicing factors. For this and other reasons stated in their review, the authors suggest that miRNAs involved in AS-related cancer should be considered therapeutic targets [109]. The interrelation between AS and epigenetic alterations is also associated with cancer-related microbiomes such as human immunodeficiency virus (HIV), Epstein-Barr virus (EBV), and human papillomavirus to mention a few. This is mainly because pre-mRNA splicing is a target for these viruses enabling them to produce proteins that inhibit cellular splicing, thus contributing to tumorigenesis. For instance, HIV uses AS to enable it to form its viral envelope transcript. It should be noted that cancers related to viral infections in SSA continue to be on the rise. The ability of oncoviruses to alter molecular processes, such as epigenetic mechanisms and AS, is their mainstay for tumorigenesis [110]. Persistent viral infection is also implicated in alterations of immune response in host organisms. Cancer-related viruses’ ability to employ mechanisms that enables them to evade immune response is connected to their ability to induce cancers. Therefore, cancer-related viruses indirectly arm cancer cells with the ability to avoid immunosurveillance. The possible impact of microbes on immune response in these settings will be discussed in more detail in the next sections.

## 12. Other Novel Therapeutic Approaches for MM

Aberrant splicing genes are a contributing factor to MM initiation and progression as indicated in the previous section. The spliceosomal gene, SF3B1, was shown to be an indicator of diffuse malignant peritoneal mesothelioma (DMPM) progression. A significant correlation between high levels of SF3B1and shorter overall and progression-free survival was found. The study showed that SF3B1 could serve as a potential novel therapeutic target for DMPM [108]. Recent advances in machine learning (ML) have allowed for advanced, accurate, and time-sensitive methods of diagnosing diseases and facilitating tailored clinical decision making. Plasma-based metabolomics and ML algorithms were investigated to assess their ability in providing the much-needed easily accessible and cost-effective diagnostic tools for mesotheliomas. Of the seven diagnostic metabolites selected based on high AUC values, taurocholic acid, tauroursodeoxycholic acid, pyrroline hydroxycarboxylic acid, and phenylalanine were upregulated in MM whilst uracil, biliverdin, and histidine were decreased. Histidine and pyrroline hydroxycarboxylic acid had the highest accuracy in detecting MM. The use of these metabolites in the diagnosis of MM could evoke improvement in the clinical prognosis of MM [111].

The Hedgehog (Hh) independent Gli activation can also be utilized as an important indicator of mesothelioma disease progression. Gli1 and 2 expression levels are highly dysregulated in MPM tissue samples. Application of Gli inhibitor suppressed cancer growth in cell cultures and xenograft models. Thus, inhibition of Gli subunits in mesothelioma can serve as novel therapeutic endeavor for this disease [112]. The circadian clock gene *BMAL1* expression levels are elevated in mesothelioma. Downregulation of *BML1* resulted in dysregulated cell cycle machinery and increased apoptosis, thus indicating its potential role as another novel therapeutic target for MPM. Studies have shown the important role of epithelial–mesenchymal transition (EMT) in cancer progression, particularly in the context of metastasis. Components of the EMT processes are constantly being studied to evaluate specific factors that can be targeted to halt cancer progression [113,114]. Lysine-specific demethylase 1 (LSD1/KDM1) is a histone-modifying enzyme that demethylates histone H3 lysine 4 and lysine 9. To study the EMT phenotype in MM, LSD1 was downregulated in the sarcomatoid MPM cell line. This downregulation promoted epithelial phenotype in this cell line and prevented transition into the EMT phenotype. These cells demonstrated sensitivity to chemotherapeutic treatment with cisplatin and induced apoptosis. Targeting LSD1 along with interconnected pathways such as the FAK–AKT–GSK3β pathway could be essential in the development of potent therapeutic strategies for MM [115].

## 13. Conclusions

Knowledge is key; thus, awareness campaigns to educate communities about the dangers of asbestos are of great importance, specifically in communities that still use asbestos products would be beneficial in regulating exposure to asbestos fibers. The inability to promptly and accurately diagnose mesothelioma makes it more challenging to find effective means of treating these patients. Therefore, MM continues to be one of the most devastating cancers with poor clinical outcomes. Patients subjected to resection surgical intervention with the hope of prolonging overall survival often succumb to the disease in less than 2 years of the initial diagnosis. The use of immunotherapy is only effective in a certain group of MM patients; however, it does not serve as the ultimate treatment solution for this disease. The reality is, that literature and the continual use of asbestos products in some countries indicate that the incidence of MM is bound to increase in the next coming years. With this in mind, the urgent discovery of effective predictive and specific therapeutic biomarkers for MM is needed.

The interconnection between the epigenetic mechanisms (including the expression of miRNAs) and alternative splicing could be key in controlling cancers. The induction of epigenetic mechanisms by asbestos fibers relates to the ability of epigenetic machinery to control AS. This interaction could be manipulated to control aberrations in AS and perhaps take away cancer-inducing microbes’ ability to take advantage of these aberrations. The involvement of microbes in cancer progression and other diseases is well documented, and ways to manipulate different molecular pathways that could finally eradicate the burden of diseases or assist in controlling drug resistance are crucial.

This review has noted the contribution of microbes in cancers closely related to mesothelioma in terms of similar mechanisms and/or relation to exposure to asbestos fibers. Chronic infections are also known to induce inflammatory responses that favor cancer progression. Microbes can induce immune responses directly or via secretions that trigger epigenetic changes. These multi-regulatory pathways hold promise in finding combinatorial therapies that will aid in the early diagnosis of MM. The lack of early diagnostic markers contributes to high mortality rates, which are mostly not recorded by WHO. The introduction of immune checkpoints has been successful in only a set group of MM patients. The components of epi-miRNAs–microbiota–immune-modulatory systems could also be manipulated for the enhancement of immunotherapeutic response or the development of efficacious therapeutic interventions.

## Figures and Tables

**Figure 1 ijms-23-08991-f001:**
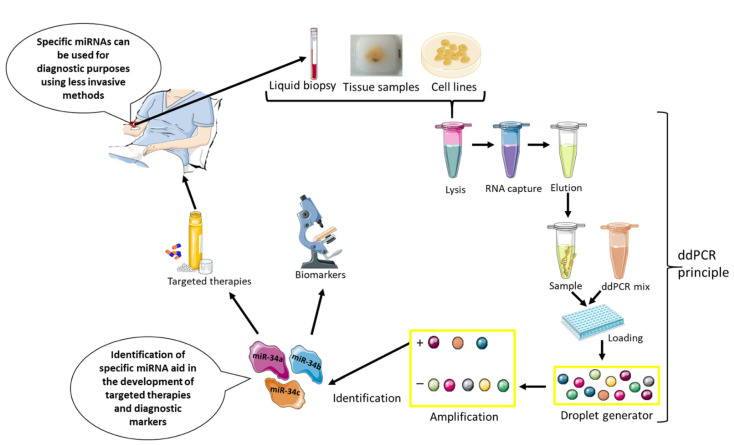
The potential application of miRNA-34a/b/c from bench to bedside and back. Multiple studies have indicated the use of miR-34 and its subgroup as potential diagnostic markers or therapeutic targets for MM. The interaction between miRNA and immune response as well as epigenetic machinery makes them highly recommendable therapeutic tools with high specificity to MM. The miRNA can be extracted from blood samples, human tissue, or MM cell lines for the identification of specific miRNAs such as miR-34 in mesothelioma, which can be processed using digital droplet PCR as depicted in the diagram. Once identified, these miRNAs can serve as diagnostic mesothelioma biomarkers. They could also be used for the development of targeted therapies that could be used alone or in combination with others. The most convenient, cost-effective, and less invasive method for the identification of miRNAs that can be used routinely as a clinically reliable diagnostic tool is a liquid biopsy.

**Figure 2 ijms-23-08991-f002:**
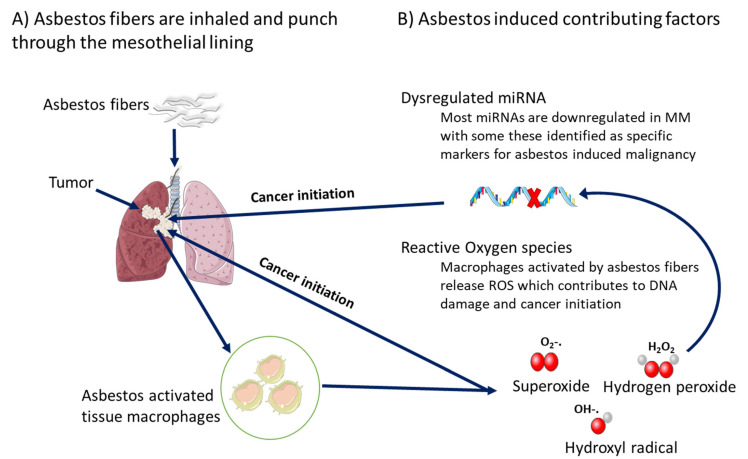
Exposure to asbestos fibers leads to cancer initiation and progression. (**A**) When inhaled or swallowed, asbestos fibers can penetrate the lining of the respiratory organs and activate cancer-inducing factors resulting in the establishment of a malignant tumor. (**B**) Asbestos-activated tissue macrophages release ROS linked to cancer initiation via aberrant gene expression (due to downregulated miRNA) and immune evasion.

**Figure 3 ijms-23-08991-f003:**
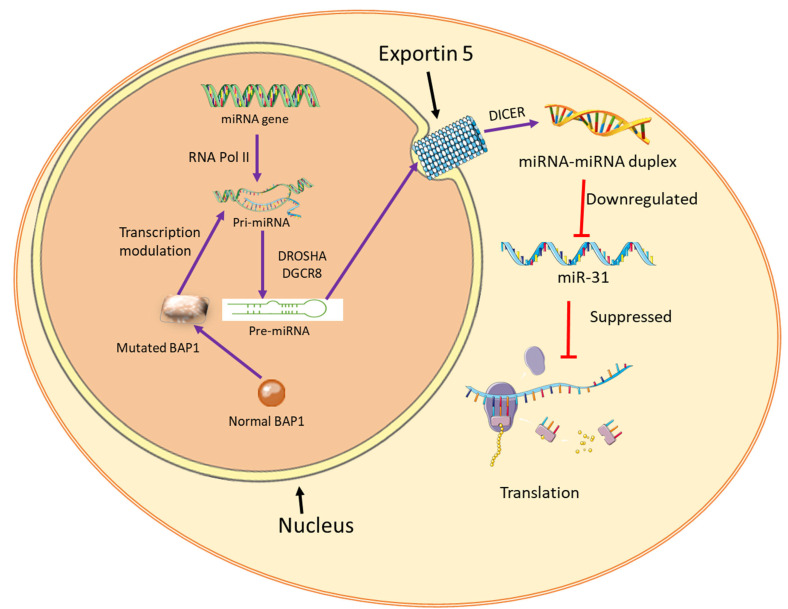
BAP1–miRNA interaction in MM.

**Figure 4 ijms-23-08991-f004:**
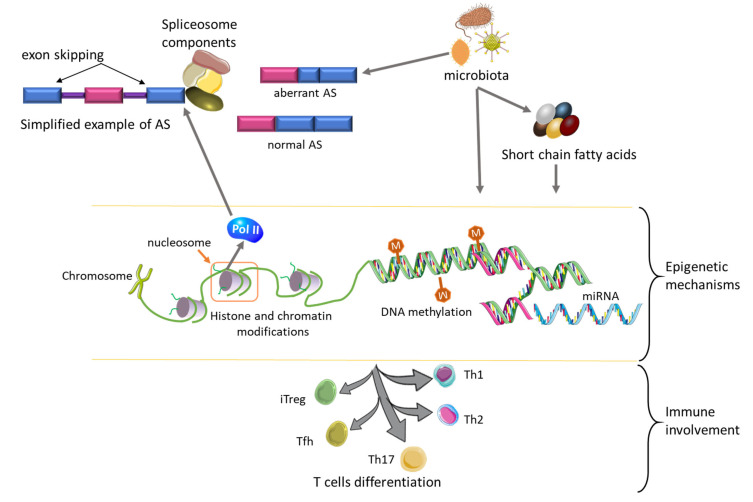
Interaction between alternative splicing, epigenetic machinery, microbiota, and immune response.

**Table 1 ijms-23-08991-t001:** miRNA signatures modulated in MM (5-year update (2017–2022)).

miRNA/s	Origin	Status in MM	References
miR-16-5p	MM cell lines exosomes	Upregulated	[36]
miR-320a	Human tissue	Downregulated	[37]
miR-548a-3p and miR-20a	Human serum	Upregulated	[38]
miR-323a-3p, miR-20b-5p and miR-101-3p	Human tissue	Downregulated	[38]
miR-137	Human tissue	Variable	[39]
	MPM cell lines	Variable	
miR-486	MPM cell lines	Downregulated	[23]
MiR-126	MM cell lines exosomes	Downregulated	[40]
miRNA-34a/b/c	Human tissue	Downregulated	[41,42,43,44]
microRNA-23b	MM cell lines	Upregulated	[45]
miR-625-3p	Human serum extracellular vesicles	Downregulated	[46]
miR-206	Human tissue	Downregulated	[24]
	Xenografts	Downregulated	
miR-18a-3p	MM cell lines	Upregulated	[47]

## Data Availability

Not applicable.

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
