# Peer review of "Microbiomes, Epigenomics, Immune Response, and Splicing Signatures Interplay: Potential Use of Combination of Regulatory Pathways as Targets for Malignant Mesothelioma"

_ijms, 2022, doi:10.3390/ijms23168991_

Round 1

Reviewer 1 Report

page 1 - line 81 - "the hearth" is repeated 

page 2 - lines 196-197- the statement "methylation profiles along with the burden of asbestos..." is questionnable, perhaps the reference (6) has been misquoted

page 2 - lines 242-244 - please, clarify the sentence 

page 3 - lines 328-332 - please clarify the sentence and add a correct reference for Wagner et al, 1960 (attached file)

page 3 - line 359 - please, give a correct reference number for Micolucci et al, 2016    

page 6 - lines 541-552 - the statement "the longer the fiber, the higher the risk of developing cancer" and the related others lack of a basis of sound evidence; please, reconsider this part

page 14 - lines 1425-1426 - a correct strategies for preventiv asbestos related disease is based upon a set of sequential measures: i) whenever possible, substitution; ii) in the remaining scenarios, working under the highest possible safety when managing asbestos - containing minerals and products; iii) in any case, managing and removing on site asbestos containing materials       

Author Response

Dear Sir/Madam,

Thank you so much for your input on the manuscript.

It is highly appreciated.

Reviewer 2 Report

In this manuscript  Setlai and coworkers review   the role of asbestos-induced epigenetic modifications, focusing on miRNAs . Authors also discuss the role of the microbiota and microbiota-epigenetic interactions on the control the innate and adaptive immunity  at the base of cancer progression and therapeutic resistance.

Malignant mesothelioma (MM)  is a  dismal disease with poor prognosis and  high mortality rates. The incidence of MM is likely to increase particularly in Low Middle Income Countries (LMICs) mostly for lack of of regulatory frameworks to control exposure to asbestos.

Although the topic of the review is interesting and timely, I think that this review need to be reorganised and  some topics require to be better discussed.

Authors have uploaded a file still containing the different revisions, making more difficult to read the manuscript. In case of a revision authors must  provide a finalised version.

Paragraph 2

Authors indicate  figures about death percentage of MM in Low Income countries  and High income countries, please give  all the data  regarding the incidence of the MM in both areas, the trends , death and survival percentages.

Authors discuss about the possible no occupational asbestos exposure. Could authors provide some references  for this type of exposure.

Instead of commenting:  This can be challenging in small communities where people usually …. Authors should discuss about the importance of the safe demolition

Paragraph 3

In this paragraph 2 authors discuss the epigenetic modifications in MM  and the possible diagnostic role , fig 1.

I think that this paragraph needs to be entirely rewritten. Authors should first discuss the role of miRNA in MM development and progression, then discuss the potential role in diagnostic, prognosis and as  therapeutic targets.

Moreover I think that before discussing miRNA in MM, authors must describe the mechanism at the base of MM , as  very briefly done in paragraph 5

Paragraph 5

Chemotherapeutic treatment of mice with tamoxifen resulted in the loss of the BAP1 gene ? please explain and provide the reference(s)

 Paragraph 8

A better general description of the interactions between microbiome and cancer should be provided together with the interactions with immune system

Since the review is focused also on the possible novel treatments  I suggest to include a specific paragraph on novel therapeutic approaches for MM

Author Response

(The authors gave the same response as above.)

Round 2

Reviewer 2 Report

Authors have revised the manuscript according with the suggestions, however in paragraph 2  there are still some unclear sentences regarding the percentage of MM related deaths.  Authors should indicate  to which  total are related these percentages: total MM cases , new cases , etc. 

page 5 line 199  I think it is better to write:  impact the trascription of genes relevant for cancer development and progression

Author Response

Dear Reviewer,

The authors would like to thank you for your continued efforts to better the manuscript. It is highly appreciated.

See attached response letter.

This manuscript is a resubmission of an earlier submission. The following is a list of the peer review reports and author responses from that submission.

Round 1

Reviewer 1 Report

1. Introduction.

MM affects the vaginal tunic of the testicles too.

Correct to mention the paper by Wagner, Sleggs & Marchand (1960), paying attention on the risk for MM from asbestos exposure both for asbestos miners and for residents near asbestos mines.

Not clearly stressed and explained a the relevance of asbestos exposure not only from asbestos mining, but from manifacturing asbestos-containing products,  asbestos-articles maintenance and asbestos-articles demolition.

Necessary to specify that the burden of asbestos - related severe pathologies comprehend not only MMs, but lung cancers, laryngeal cancers, asbestosis and others too.

It is false that the "current limitations" of histology and immunohistochemistry as diagnostic tools for MMs produce a lack of "sensitivity and specificity including variations modelled by operator technical skills".    

The alternative diagnostic tools proposed by the Authors have being in effect studied for years and their possible practical utility seems to be restricted, still today, to attempts of an early diagnosis of initial cases not revealable by radiology and other classical approches.      

9. Surprisingly, the paper miss of any mention of the utility and necessity of prevention by the control of asbestos occupational exposures and of asbestos general poluution, pointing the attention just upon the early diagnosis of the MM as an assumed opportunity of ameliorationg the prognosis.    

Reviewer 2 Report

This review article considers the various roles of the microbiome, epigenomic and splicing signature towards the immune response and therapeutic targets in mesothelioma. A clear gap in knowledge and clinical need is communicated, although it is suggested that the authors approach this from a more focused standpoint- to cover sufficient detail and depth- both pathogenesis (asbestos fibres) and treatments are considered (perhaps focus on either development or treatment?). In my opinion, the immunogenic details expected (phenotype/ complexity signalling/TME, M1 vs. M2/NOTCH) are somewhat lacking- so perhaps consider focussing on the epi-regulatory roles of MicroRNAs in MM (given the recent interest of the new Hallmark of Cancer: New Dimensions review). With comparisons to other cancers, perhaps compared MM to the upper respiratory tract and oral microbiome. Specific microRNAs may also have diagnostic utility in differentiating patients with malignant pleural mesothelioma from benign asbestos-related pleural effusion.

Several recent reviews already cover this content in detail e.g. RNA editing in mesothelioma (and genetic and epigenetic alterations) in the context of current therapies. The authors may wish to focus on the relevant microRNAs functional studies of this cancer, more reviews would be welcomed in this area.

The figures and images need attention, they are difficult to interpret and some lack the scientific detail that would be expected in a review (e.g the cell cycle diagram in Figure 4). The paragraphs need shortening and some attention is needed to ensure the English language is appropriate and understandable. More detail needed in the abstract and conclusion. However, this is a very interesting review and I think a more focussed and pragmatic approach (with new diagrams) would be welcomed by the mesothelioma community.
